# The Mechanism of Dendrite Formation in a Solid-State Transformation of High Aluminum Fe-Al Alloys

**DOI:** 10.3390/ma16072691

**Published:** 2023-03-28

**Authors:** Haodong Yang, Yifan Zhang, An Zhang, Frank Stein, Zhengbing Xu, Zhichao Tang, Dangjing Ren, Jianmin Zeng

**Affiliations:** 1State Key Laboratory of Featured Metal Materials and Life-Cycle Safety for Composite Structures, Guangxi University, Nanning 530004, China; 2Max-Planck-Institut für Eisenforschung GmbH, Max-Planck-Str. 1, 40237 Düsseldorf, Germany; 3Key Laboratory of High-Performance Structural Materials and Thermo-Surface Processing, Guangxi University, Education Department of Guangxi Zhuang Autonomous Region, Nanning 530004, China; 4Centre of Ecological Collaborative Innovation for Aluminium Industry in Guangxi, Nanning 530004, China

**Keywords:** Fe-Al alloy, solid-state dendrite, in-situ observation, phase transformations

## Abstract

The mechanism of solid-state dendrite formation in high-aluminum Fe-Al alloys is not clear. Applying an in-situ observation technique, the real-time formation and growth of FeAl solid-state dendrites during the eutectoid decomposition of the high-temperature phase Fe_5_Al_8_ is visualized. In-situ experiments by HT-CSLM reveal that proeutectoid FeAl usually does not preferentially nucleate at grain boundaries regardless of rapid or slow cooling conditions. The critical radii for generating morphological instability are 1.2 μm and 0.9 μm for slow and rapid cooling, respectively. The morphology after both slow and rapid cooling exhibits dendrites, while there are differences in the size and critical instability radius *R*_c_, which are attributed to the different supersaturation *S* and the number of protrusions *l.* The combination of crystallographic and thermodynamic analysis indicates that solid-state dendrites only exist on the hypoeutectoid side in high-aluminum Fe-Al alloys. A large number of lattice defects in the parent phase provides an additional driving force for nucleation, leading to coherent nucleation from the interior of the parent phase grains based on the orientation relationship {$\bar{3}3$0}_Fe5Al8_//{$\bar{1}$10}_FeAl_, <11$\bar{1}$>_Fe5Al8_//<11$\bar{1}$>_FeAl_. The maximum release of misfit strain energy leads to the preferential growth of the primary arm of the nucleus along <11$\bar{1}$> {$\bar{1}$10}. During the rapid cooling process, a large supersaturation is induced in the matrix, driving the Al atoms to undergo unstable uphill diffusion and causing variations in the concentration gradient as well as generating constitutional undercooling, ultimately leading to morphological instability and the growth of secondary arms.

## 1. Introduction

Fe-Al based alloys are a promising high temperature structural material due to their excellent high temperature oxidation resistance, low density, and high strength [1,2,3,4]. Although the Fe-Al alloys containing up to 50 at.% Al have been well studied, there is a lack of research for aluminum contents exceeding 50 at.%. In the composition range of around 60 at.% Al, the unquenchable high-temperature phase Fe_5_Al_8_ exists [5,6,7,8]. It was found [5] that this phase, which decomposes eutectoidally at 1095 °C, precipitated an uncommon solid-state dendritic FeAl phase + lamellar microstructure (FeAl+FeAl_2_) after its eutectoid decomposition.

It is well known that the formation of dendritic morphology during solidification is affected by constitutional undercooling [9], interfacial energy anisotropy, interfacial instabilization [10,11], solute redistribution, convection, etc. In the case of solid-state transformations, these conditions are not identical; precipitated morphology especially is not affected by solute redistribution or convection, and the role of elastic strain cannot be neglected in solid-state phase transformations [12,13,14,15,16]. Studies on the formation of solid-state dendrites in Ni-based alloys [13,17,18,19,20,21,22,23,24,25] have shown that there are differences in the morphology and size of solid-solution dendrites and intermetallic compound dendrites due to different atomic diffusion mechanisms and crystal structures, which affect the evolution of dendrites. At present, the mechanism of nucleation, growth, and morphological evolution of proeutectoid FeAl in dendritic morphology in the solid state has not been reported. In explaining the mechanism of solid-state dendrite formation, researchers usually preserve the morphology of dendrites at different stages by quenching for ex-situ studies, which is not conducive to obtaining the characteristics of real-time nucleation and morphological evolution of dendrites, especially in the case of rapid decomposition reactions.

Therefore, in this work, the method of in-situ high temperature confocal scanning laser microscopy (HT-CSLM) is applied to study the real-time evolution of dendrites in a rapid eutectoid reaction during continuous cooling of Fe-Al alloys. In addition, the orientation characteristics of solid-state dendrites were investigated through ex situ characterization by electron backscatter diffraction (EBSD) aiming at an improved understanding of solid-state dendrite nucleation and growth.

## 2. Materials and Methods

A series of Fe-Al alloys of 200g each were prepared by vacuum induction melting of Fe (99.95 wt.% purity) and Al (99.999 wt.% purity). The chemical compositions of the alloys were determined as Fe-59.9Al, Fe-61.1Al, and Fe-62.7Al (in at.%) by electron probe microanalysis (EPMA, JXA-8100, JEOL, Tokyo, Japan). Microstructures were studied by scanning electron microscopy (SEM, JSM-6490, JEOL, Japan) and (Sigma 300, Zeiss, Oberkochen, Germany) equipped with an energy dispersive spectroscope (EDS, Xplore 30, OXFORD, Oxford, UK). In situ observations were carried out using HT-CSLM (VL2000DX-SVF18SP, Yonekura, Japan). Fe-59.9Al alloy samples sized as Φ 7 × 3 mm cylinders were placed in a halogen-heated furnace together with pure Ti flakes as the deoxidizer; the chamber vacuum was pumped to 10^−2^ Pa and then backfilled with ultra-purity argon before heating. The heat treatment schemes of the samples are shown in Figure 1. Images were captured at 15 fps and the entire phase transformation was recorded in real-time. The orientation of the quenched Fe-59.9Al sample was analyzed using SEM (Quanta FEG 450, FEI, Hillsboro, OR, USA) equipped with an EBSD detector (DigiView, EDAX, Mahwah, NJ, USA) with a step size of 0.5 μm to obtain an index image of 350 × 270 μm. Points in the FeAl phase with confidence indices (CIs) ≥ 0.3 were screened. All EBSD data were analyzed using OimA 7.0 software (TSL, Draper, UT, USA).

## 3. Results

### 3.1. Microstructural Comparison of Fe-Al Alloys with Different Compositions and Heat Treatments

Figure 2 shows a comparison of the Fe-59.9Al, Fe-61.1Al, and Fe-62.7Al alloys in as-cast conditions, after furnace cooling (F.C.) and after brine quenching (B.Q.), respectively. From the comparison of the SEM images above, it can be seen that the dendritic morphology (proeutectoid FeAl) is only found in the microstructure of the Fe-59.9Al hypoeutectoid alloy, while the proeutectoid FeAl_2_ phase of Fe-61.1Al and Fe-62.7Al hypereutectoid alloys was striped or needle-like.

The analysis of the crystal structures of Fe_5_Al_8_, FeAl, and FeAl_2_ showed that FeAl (*Pm*3*m*, No. 221, *cP*2) and Fe_5_Al_8_ (*I*43*m*, No. 217, *cI*52) are both cubic structures [5]. The lattice parameter of Fe_5_Al_8_ is about three times that of FeAl and the lattice mismatch is only 0.7% (cf. Table 1), while the triclinic FeAl_2_ phase (*P*1, No. 2, *aP*19; *a* = 4.9548(3) Å, *b* = 6.5661(6) Å, c = 8.925(1) Å, α = 91.513(7)°, β = 73.446(7)°, γ = 97.058(7)° at 1080 °C [5]) is completely incoherent with Fe_5_Al_8_. The comparison shows that solid-state dendrites appear to occur only in systems where the parent and precipitated phases are of similar crystal structure. This assumption is supported by observations of dendritic solid-state precipitates in other systems (see Table 1 which is based on a table shown in [16]), confirming the results of Malcolm and Purdy [26] and Husain et al. [16]. It should be mentioned that in these systems, the parent phase is usually a complex Hume-Rothery-type superstructure phase, while in Fe-Al alloys the opposite is true. The effect of this is discussed later.

### 3.2. In Situ Observation of the Fe-59.9Al Alloy under Different Cooling Rates

To visualize the nucleation, growth, and evolution of the dendritic microstructure, in situ experiments with continuous cooling at different cooling rates were designed, as shown in Figure 1. Scheme I simulates the furnace cooling process; its in situ microstructural evolution is shown in Figure 3 (see also Appendix A, which reveals the evolution in detail). The non-equilibrium proeutectoid transformation of Fe_5_Al_8_ → FeAl occurs in Fe-59.9Al at 1098.6 °C with an undercooling of about 25.7 °C. The proeutectoid FeAl phase prefers to nucleate at oxide inclusions on the free surface because of the lower energy barrier compared to grain boundary nucleation. Morphological instability occurs when the semi-short axis is about 1.2 μm. The final length of the dendrites is about 100–150 μm. Under non-equilibrium cooling conditions, the rapid eutectoid decomposition of Fe_5_Al_8_ → FeAl + FeAl_2_ occurs at about 1043.5 °C, below the equilibrium phase transformation point (since the eutectic FeAl+FeAl_2_ lamellae are very thin, the transformation can be determined from the sudden light–dark contrast observed in the in situ experiment, as shown in Figure 3d).

Scheme II simulates the quenching process of Stein et al. [5] (air cooling+quenching), and its in situ microstructure evolution is shown in Figure 4 (see also Appendix A). The proeutectoid transformation of Fe_5_Al_8_ → FeAl occurs at 1050.6 °C. The undercooling (about 73.6 °C) is higher compared to the slow-cooling case. Proeutectoid FeAl prefers to nucleate on the free surface, which indicates that the nucleation barrier is extremely low, even lower than that of inclusions and grain boundary nucleation [31], and it showed an ellipsoidal shape with an obvious tendency to preferential growth.

It should be mentioned that, for the nucleation and evolution processes in the in situ experiments, there are similarities and differences between those occurring at the free surface and in the bulk [32]. It is certain that the free surface is the most effective nucleation site in terms of nuclei number and nucleation priority. Secondly, in the case of slow cooling, oxide inclusions nucleation occurs only at the surface. While in the case of rapid cooling, there is a correlation between the condition of the free surface and the condition in the bulk as far as nucleation and growth are concerned. In the present system, a large amount of precipitates from the interior of the grain indicates that lattice defects (misfit lattices; see later Section 4) contribute significantly to nucleation, as shown in Figure 5a, but of course these defects can also be present on the free surface. This is different from δ-ferrite → γ-austenite which usually occurs at grain boundaries [32,33] because their coherent relationships and defects are different from this system.

Therefore, it is believed that the large number of dendrites from the interior of the parent grain is related to the nucleation of lattice defects. In addition, this preferential growth implies the existence of a certain orientation relationship between the two phases (see later Section 4). Morphology instability occurs when the semi-short axis is about 0.9 μm. It can be observed from Figure 4d that the secondary arms of the FeAl phase grow at an angle of about 55.5° on the lower side of the long axis, while the secondary arms of the FeAl phase in Figure 4e grow at about 55.7° on the upper side of the long axis. The secondary arm spacings are about 3–5 μm, and the secondary arms of each dendrite appear almost exclusively on one side. Meanwhile, some of the dendrites also grew up with obvious triple arms. In addition, dendrite growth will be suppressed or bent due to elastic effects between neighboring precipitates and overlapping diffusion fields which results in rapid removal of supersaturation. The final length of the dendrites is relatively small, about 50–100 μm. It can be seen that the difference of supersaturation resulting from different cooling rates plays an important role in the morphology evolution of the solid-state dendrites.

### 3.3. Orientation Characteristics of FeAl Dendrites

SEM and EBSD were used to reveal the dendritic morphology and orientation of the Fe-59.9Al samples after the in situ experiments of Scheme II as shown in Figure 5. It can be seen that three different orientation regions form typical triangular large grain boundaries that originate from the parent Fe_5_Al_8_ single-phase state. The white proeutectoid FeAl precipitates in the interior of grains have dendritic morphology and show preferential orientations (see Figure 5a). The proeutectoid FeAl precipitates at the grain boundaries are irregular: the flat side is an incoherent interface and the other protrusion side is consistent with the intragranular FeAl orientation [26]. The dark regions are finely lamellar eutectoid FeAl+FeAl_2_ mixtures generated from the Fe_5_Al_8_ eutectoid decomposition. Figure 5b shows that the orientation and morphology of the precipitated phases within the same parent grains tend to be consistent. The green phases are FeAl dendrites, and the blue–purple and yellow–green phases are identified as dendritic oblique cross-sections by orientation analysis.

According to the pole figures of Figure 6a,b, the green phase with the largest grain size and the longest long axis in Figure 5b has a {110} texture in the center of the pole figure and a {111} texture on the outermost circle of the pole figure, indicating that the main axis of the dendrite is parallel to {110} and has a preferred orientation <$11\bar{1}$>.

In addition, the angle between the traces <$11\bar{1}$>, <00$\bar{1}$> of Figure 6b,c and the angle between the primary and secondary arms of the dendrite of Figure 6d both tend to be 55.7°. This preferred growth implies that there must be some kind of orientation relationship between the proeutectoid FeAl and the parent phase. Moreover, the angle seems to be related to the competing effects between supersaturation, interfacial energy anisotropy, elastic anisotropy, etc., as well as the elastic interaction between neighboring precipitated phases and the overlapping diffusion field [34], as discussed below.

## 4. Discussion

It is well known that there is no elastic stress during solidification, and the surface energy anisotropy and supersaturation determine the morphology evolution of precipitates. However, in solid-state phase transformations, we must consider the role of elastic strain energy anisotropy. Therefore, for the observed Fe-Al solid-state dendrites, the formation conditions and mechanisms are as follows.

### 4.1. Nucleation Mechanism of the FeAl Precipitates

The total interfacial energy of the new phase is proportional to the surface area and the elastic strain energy is proportional to the volume of that. When the new phase size is very small, the total interfacial energy dominates the energy barrier and the interfacial free energy anisotropy determines the growth direction. Usually, the nucleation of the precipitated phase keeps a certain orientation relationship with the parent phase, and good atomic matching at the interface usually occurs in the closed-packed plane of the two phases, which minimizes the interfacial energy [35,36]. It is known that the closed-packed system of the body-centered cubic structure is {110}<111>, and preferential growth usually occurs on the closed-packed plane. Unfortunately, the high temperature phase (Fe_5_Al_8_) cannot be retained at room temperature [5]. Therefore, the orientation relationship for Fe_5_Al_8_/FeAl can only be inferred by the edge-to-edge matching (E2EM) model [35] and the directed growth model [37]. Scherf [38] inferred that Fe_5_Al_8_ and FeAl have a cubic-to-cubic orientation relationship [111]_Fe5Al8_//[111]FeAl; the Cu-Zn alloy [39] β/γ has the orientation relationship {110}_β_//{110}_γ_, <111>_β_//<111>_γ_, the crystal structure of β/γ is the same as FeAl/Fe_5_Al_8_, and this orientation relationship is expected to exist. According to the E2EM model [35], the most favorable case of nucleation occurs in closed-packed rows and planes with minimum mismatch between the two phases, based on the principle of minimum interfacial energy. The mismatches are defined as follows:(1)$$f_{r}=\frac{\left|r_{M}-r_{P}\right|}{r_{P}}$$
(2)$$f_{d}=\frac{\left|d_{M}-d_{P}\right|}{d_{P}}$$
where *ƒ*_r_ is the interatomic spacing misfit between a pair of closed-packed rows, *r*_M_ and *r*_P_ are the interatomic spacings along the matching direction in the parent and precipitated phases, respectively, *ƒ*_d_ is the interplanar spacing mismatch between a pair of closed-packed planes that contain the rows, and *d*_M_ and *d*_P_ are the interplanar spacing of the close-packed plane in the parent and precipitated phases, respectively. The basic features of the E2EM model of Fe_5_Al_8_/FeAl are shown schematically in Figure 7e. The *ƒ*_r_ of <11$\bar{1}$>_Fe5Al8_ with <11$\bar{1}$>_FeAl_ is 0.67% in the close-packed atom rows and *ƒ*_d_ of {$\bar{3}$30}_Fe5Al8_ with {$\bar{1}$10}_FeAl_ is 0.67% on the close-packed plane as was calculated from the lattice parameters reported in [5]. Fitting rows are <11$\bar{1}$>_Fe5Al8_//<11$\bar{1}$>_FeAl_, and matching planes are {$\bar{3}$30}_Fe5Al8_//{$\bar{1}$10}_FeAl_ as shown in Figure 7a–c. Therefore, combined with the preferred growth observed in Figure 5 and Figure 6, it can be inferred that nucleation is based on the orientation relationship {$\bar{3}$30}_Fe5Al8_//{$\bar{1}$10}_FeAl_, <11$\bar{1}$>_Fe5Al8_//<11$\bar{1}$>_FeAl_.

It should be mentioned that the Fe_5_Al_8_ phase was found to have a body-centered cubic structure of the Hume–Rothery Cu_5_Zn_8_ type (I4$\bar{3}$m (No.217), *Z* = 4, cI52) and 52 atoms and missing atoms at the corners and the center in the unit cell [5]. In this case, as in Figure 7d,e, the close-packed plane ($\bar{3}$30)_Fe5Al8_, ($\frac{\bar{3}}{2}\frac{3}{2}\;$0)_Fe5Al8_ is coherent with ($\bar{1}$10)_FeAl_, whereas of the low index plane ($\bar{1}$10)_Fe5Al8_ of unit cell contains 5/25 misfitted atoms. These misfitted atoms are considered as lattice defects (they maybe form dislocations or vacancies) and have a significant impact on the nucleation position. Combined with a large number of precipitates from the interior of parent grains in Figure 5, it is inferred that the precipitates are nucleated by adhering to a large number of lattice defects, which is due to the ability of lattice defects to provide sufficient driving force [31].

### 4.2. Morphological Evolution Mechanism of the FeAl Precipitates

When some proeutectoid FeAl nuclei grow to a certain size with increasing supersaturation, the elastic strain energy gradually dominates [31]. The morphological evolution is related to the reduction of elastic strain energy caused by the mismatch of the parent phase lattice and the elastic anisotropy [40].

As in Figure 7d,e, the center of the parent phase single cell ($\bar{1}$10)_Fe5Al8_ has a vacancy-like defect relative to the precipitated phase ($\bar{1}$10)_FeAl_ and the neighboring lattice distortion maximum along the <11$\bar{1}$> direction, when the highest stored elastic strain energy density is available. As the precipitated phase grows, the ‘a’ atom is filled by the nearest ‘D’ atom, so the ‘b’ atom is pushed and thus coherent with the ‘B’ atom. In this case, the lattices of [11$\bar{1}$]_FeAl_ are expanded relative to those of the [11$\bar{1}$]_Fe5Al8_. Therefore, the preferred growth along {$\bar{1}$10}<11$\bar{1}$>_FeAl_ can minimize the elastic strain energy which may be the main reason for the preferential growth of dendrites along <11$\bar{1}$> as shown in Figure 5 and Figure 6.

In addition, according to the directed growth model [37], the direction of the maximum growth rate of the stable phase is the direction of the highest elastic strain energy density stored in the metastable matrix, i.e., the direction of the maximum Young’s modulus (MxYMD) at the coherent interface with the lowest interfacial energy. Based on this energy condition, it is also possible to derive the orientation relationship between the precipitate and the matrix. FeAl is highly anisotropic with an elastic anisotropy constant *A*_FeAl_ = 2.51 [41]. However, the elastic anisotropy constant of the matrix phase Fe_5_Al_8_, *A*_Fe5Al8_ is equal to −0.72 < 1, so it cannot be preserved stably to room temperature [8,42,43]. Given the compliance tensors *S*_11_ = 0.0067 GPa^−1^, *S*_12_ = −0.0024 GPa^−1^, and *S*_44_ = 0.0073 GPa^−1^, FeAl Young’s moduli values *E*(_hkl_) on the crystallographic orientation (hkl) are calculated according to Equation (3) [44]:(3)$$E_{\left(hkl\right)}{}^{-1}=S_{11}\;-\;2(S_{11}\;-\;S_{12}\;-\;0.5S_{44}\;)\left(\frac{h^{2}k^{2}+k^{2}l^{2}+h^{2}l^{2}}{{\left(h^{2}+k^{2}+l^{2}\right)}^{2}}\right)$$

The corresponding 3D orientation distribution is plotted by the open-source online application (ELATE) [45] as shown in Figure 8, which shows that the <11$\bar{1}$> direction of FeAl is MxYMD. Therefore, the preferred growth direction <11$\bar{1}$>_FeAl_ is reasonable. In summary, it was further justified that this orientation relationship ($\bar{3}$30)_Fe5Al8_//($\bar{1}$10)_FeAl_, <11$\bar{1}$>_Fe5Al8_//<11$\bar{1}$>_FeAl_ is reasonable. The morphological evolution of the FeAl phase indicates that it is more influenced by elastic strain anisotropy in the subsequent growth process.

The secondary arms tending to <00$\bar{1}$> can be observed in Figure 6d; its formation and evolution are closely related to the interfacial stability. Yoo et al. [19,46] showed that the dependence of morphological instability on supersaturation remains valid if sufficient supersaturation is present in the matrix. According to the analysis of Mullins and Sekerka [10], the critical instability radius *R*_c_ of precipitate can be described by Equation (4):(4)$$R_{c}\left(l\right)=\left[\frac{1}{2}\left(l+1\right)\left(l+2\right)+1\right]R^{*}$$
(5)$$R^{*}=\frac{2\Gamma_{D}}{S}$$
where *l* is the number of protrusions, *R** is the critical radius of the nucleation, *Γ_D_* is the capillary constant, and *S* is the relative supersaturation. The solute diffusion at high temperature is mainly Al atoms because of its higher diffusion coefficient [47]. During the continuous cooling processing, the high undercooling causes a sharp increase in supersaturation relative to that of the equilibrium state when *R** decreases, and these particles rapidly grow to the critical instability size *R*_c_. Slight interfacial energy anisotropy provides a steady distorting force during the development of the perturbations. The formation of perturbations leads to interface instability and causes slight protrusions eventually [26]. Meanwhile, rapid cooling reduces the uphill diffusion rate of Al atoms and forms an Al-rich zone at the phase boundary. Meanwhile, the Fe atom content remains stable in the secondary arm by rapid exchange with vacancies and forms a Fe-depleted zone at the phase boundary, as shown in Figure 9a. In addition, changes in concentration gradients produce constitutional undercooling and promote the development of solid-state dendrites which have been reported [48]. In addition, when the size of secondary arms meets their critical instability radius *R*_c_, higher orders of arms will start to grow and the dendritic morphology of FeAl phase will form, as shown in Figure 9b and Figure 4c in the in situ experiment. The evolution model of FeAl solid-state dendrite is shown in Figure 9c.

Through the in situ experiments in Figure 3 and Figure 4, it is observed that the morphology after both slow and rapid cooling exhibits dendrites, while there are differences in the size and critical instability radius *R*_c_, which are attributed to the different supersaturation *S* and the number of protrusions *l*. The rapid cooling condition makes the supersaturation and the constitutional undercooling increase while *R** decreases and grows stronger dendrites relative to the slow cooling condition. In addition, as the precipitates continue to grow to very small spacings, this will cause their diffusion fields to overlap and the effective diffusivity will decrease [49] and the supersaturation in the matrix will decrease rapidly. The dendrites continue to grow and stop before the start of the next stage of eutectoid decomposition.

### 4.3. The Expected Formation Conditions of Solid-State Dendrites

Usually, most solid-state phase transformations are unable to form dendrites because the parent phase has a different crystal structure from the precipitating phase or the lattice is not well matched in three dimensions. Therefore, they preferentially undergo nucleation at the grain boundaries to reduce the nucleation energy barrier and cannot form regular dendrites. Alternatively, the supersaturation of the matrix is so low and the precipitated phase does not reach the critically unstable size so that elastic strain and interfacial instability effects cannot work effectively.

Based on the experimental results and analysis, we propose two necessary conditions for the formation of solid-state dendrites.

(1)Crystallographic conditions: crystallographic similarity, including similar crystal structures, 3D matching, and suitable orientation relationships, which will ensure that the new phase can preferentially precipitate coherently in the interior of the parent grain, which is the primary condition for exhibiting regular dendrite precipitate, while the grain boundary precipitate is almost irregular.(2)Thermodynamic conditions: appropriatesupersaturation, low interfacial energy, and coherent strain energy as well as elastic interactions to regulate the characteristic evolution of dendritic morphology.

## 5. Conclusions

In this work, a combination of in situ observations and crystallographic and thermodynamic analysis was applied to systematically investigate the formation and evolution mechanism of FeAl dendrites during continuous cooling through the solid-state eutectoid decomposition reaction of the high-temperature phase Fe_5_Al_8_. The main conclusions are as follows:
(1)The solid-state dendrites exist only on the hypoeutectoid side in high-aluminum Fe-Al alloys, due to the crystallographic similarity of Fe_5_Al_8_ and FeAl.(2)In situ observations by HT-CSLM reveal that proeutectoid FeAl preferentially nucleates at the oxide inclusions during slow cooling, while during rapid cooling, the free surface and lattice defects provide additional driving force for nucleation. The critical radii for generating morphological instability are 1.2 μm and 0.9 μm for slow and rapid cooling, respectively, as obtained from the in situ experiments. The morphology after both slow and rapid cooling exhibits dendrites, while there are differences in size and critical instability radius *R*_c_, which are attributed to the different supersaturation *S* and the number of protrusions *l*.(3)A large number of lattice defects in the parent phase provides an additional driving force for nucleation, leading to nucleation from the interior of parent phase grains based on the orientation relationship {$\bar{3}3$0}_Fe5Al8_//{$\bar{1}$10}_FeAl_, <11$\bar{1}$>_Fe5Al8_//<11$\bar{1}$>_FeAl_. The maximum release of misfit strain energy leads to the growth of the primary arm of the nucleus along <11$\bar{1}$>{$\bar{1}$10}. During the rapid cooling process, a large supersaturation is induced in the matrix, driving the aluminum atoms to undergo unstable uphill diffusion and causing variations in the concentration gradient as well as the constitutional undercooling, ultimately leading to morphological instability and the growth of secondary arms.

## Figures and Tables

**Figure 1 materials-16-02691-f001:**
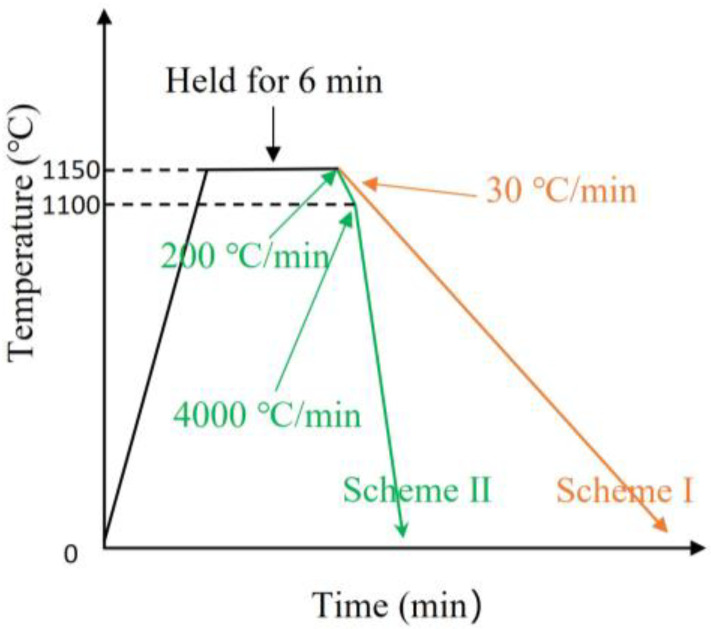
The heat treatment schemes of the in situ experiments.

**Figure 2 materials-16-02691-f002:**
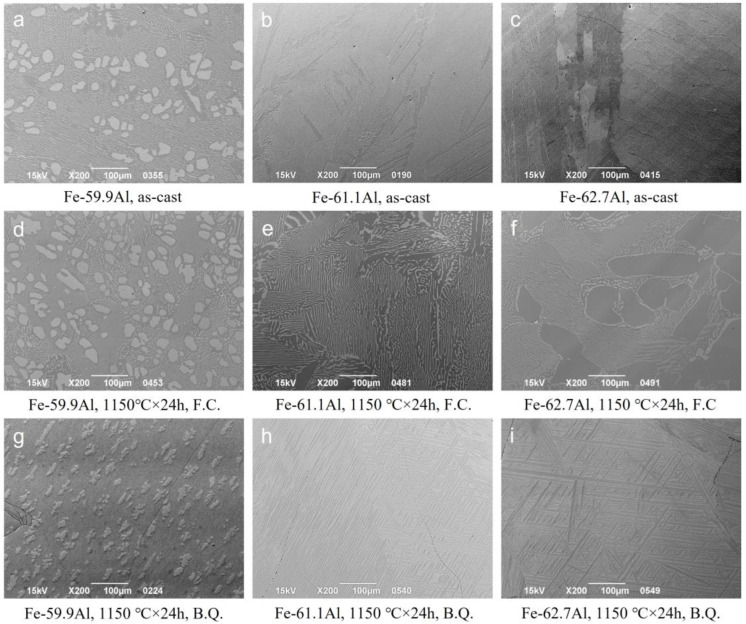
Selected SEM images of the investigated Fe-Al alloys (compositions in at.%).

**Figure 3 materials-16-02691-f003:**
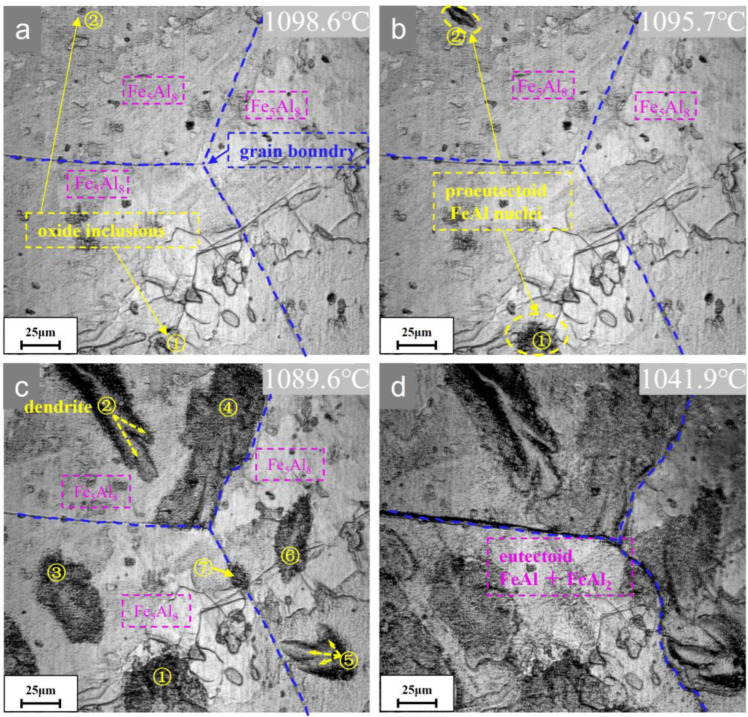
Microstructural evolution of Fe-59.9Al alloy during cooling according to Scheme I.

**Figure 4 materials-16-02691-f004:**
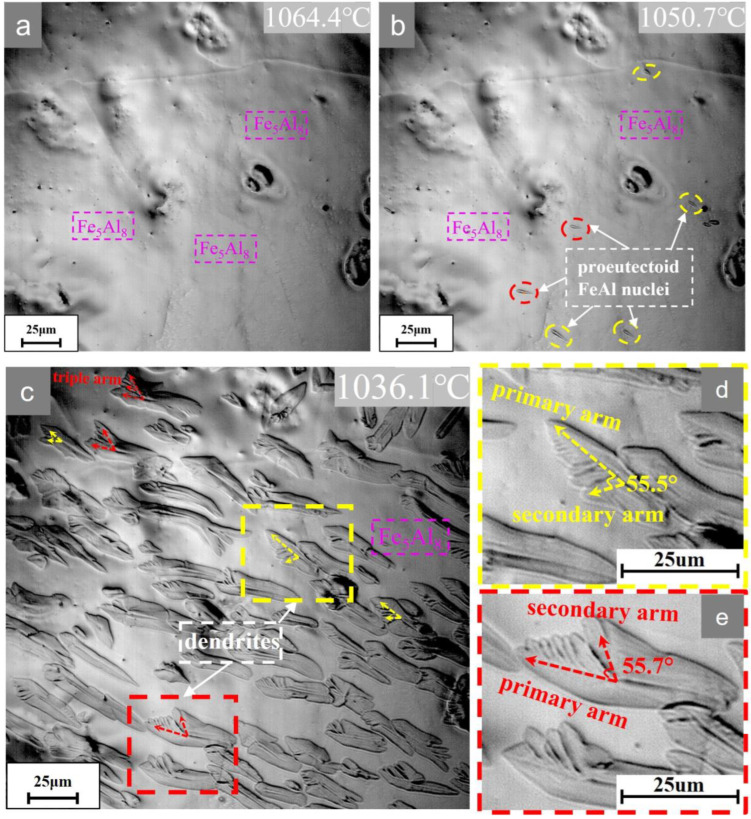
Microstructural evolution of Fe-59.9Al alloy during cooling according to Scheme II.

**Figure 5 materials-16-02691-f005:**
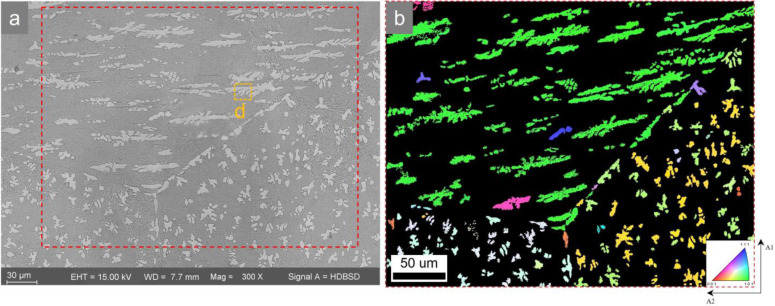
(**a**) the SEM image after rapid cooling (backscattering mode). (**b**) Full Euler image of the area marked by the red box in (**a**).

**Figure 6 materials-16-02691-f006:**
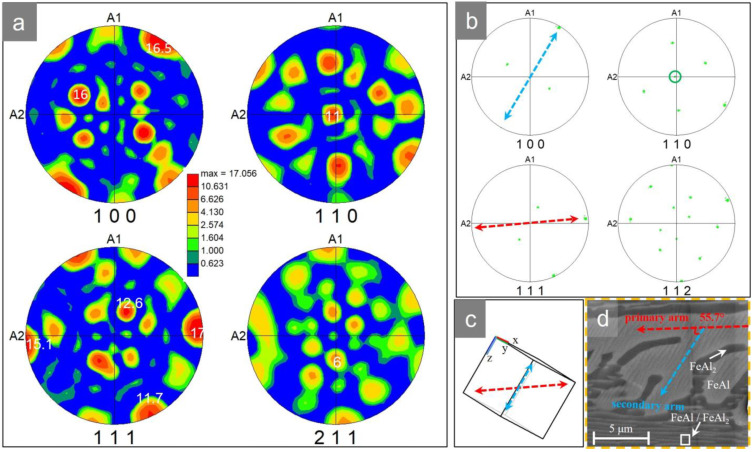
(**a**) Pole figures in Figure 5b. (**b**) Scattered pole figures of long green dendrites in Figure 5b. (**c**) Crystal orientation diagram of the secondary arm of the area ‘d’ marked in yellow in Figure 5a. (**d**) SEM image of yellow-marked area ‘d’ in Figure 5a.

**Figure 7 materials-16-02691-f007:**
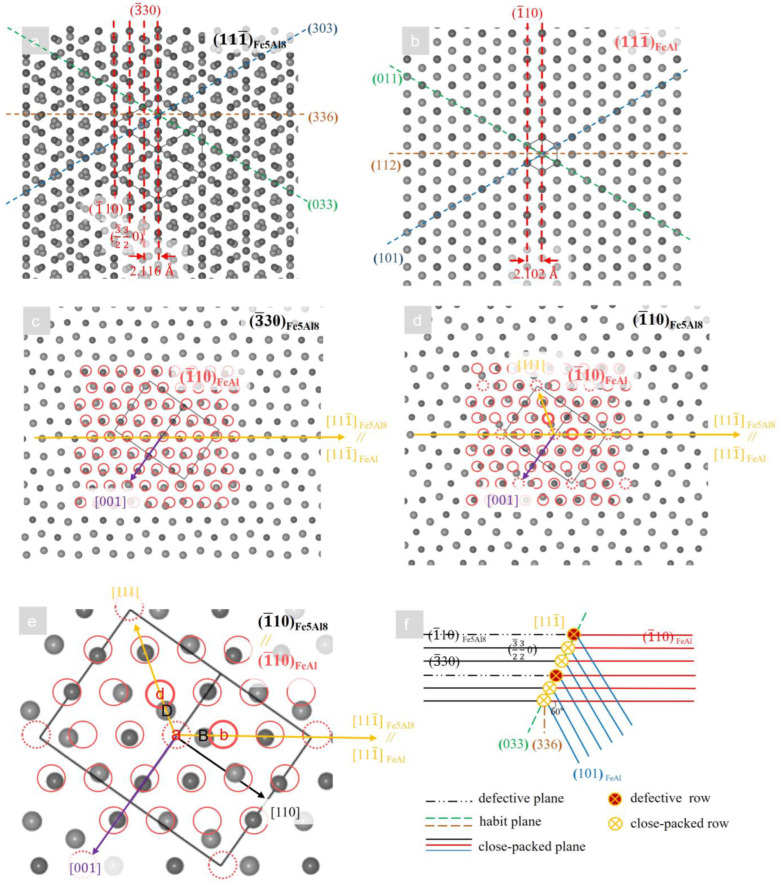
(**a**,**b**) are schematic drawings of the atomic arrangements of (11$\bar{1}$)_Fe5Al8_ and (11$\bar{1}$)_FeAl_. (**c**,**d**) are matching situations of Fe_5_Al_8_ and FeAl atoms. (**e**) Local enlargement in ‘d’. (**f**) is a schematic view of the essential features of the E2EM model.

**Figure 8 materials-16-02691-f008:**
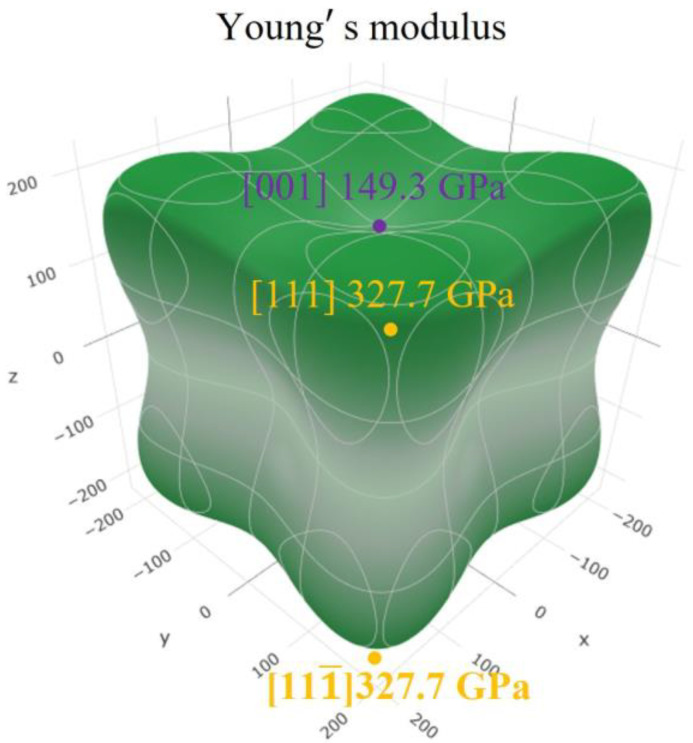
Three-dimensional orientation map of FeAl Young’s modulus.

**Figure 9 materials-16-02691-f009:**
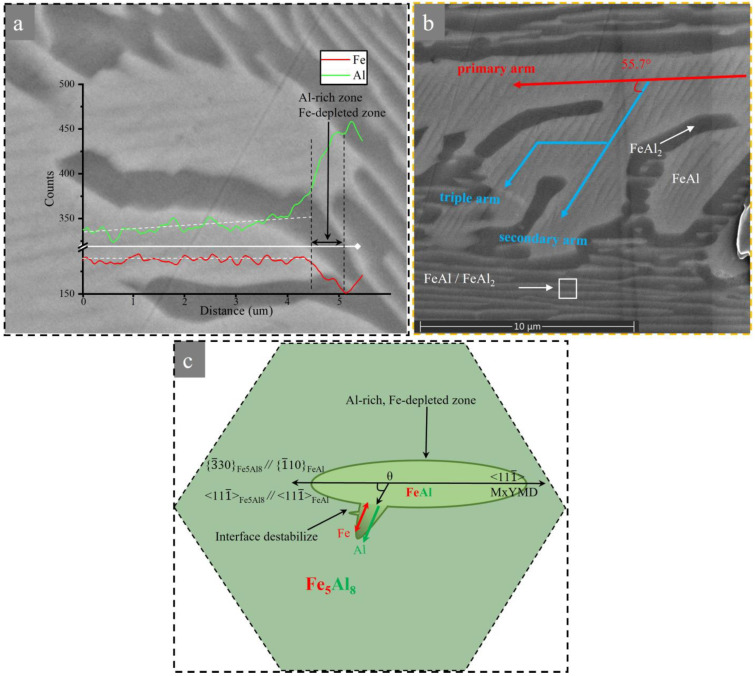
(**a**) SEM-EDS line scan of the secondary arm. (**b**) SEM image of the area ‘d’ in Figure 5a. (**c**) The evolution model of FeAl solid-state dendrite.

**Table 1 materials-16-02691-t001:** Crystallographic features of the parent phase and the precipitate in binary alloys in which dendritic morphology is confirmed [16] compared with the present Fe-Al phases (last line).

Alloy System	Parent Phase			Precipitate			Mismatch (%)
	Phase	Structure Type	Lattice Parameter (Å)	Phase	Structure Type	Lattice Parameter (Å)	
Cu-Al [27]	Cu_3_Al (β)	*A*2	2.9564	Cu_9_Al_4_ (γ_2_)	*D*8_3_	8.7068	1.8
Cu-Sn [28]	Cu_3_Sn (γ)	*D*0_3_	6.0605	Cu_41_Sn_11_ (δ)	-	17.980	1.1
Ni-Zn [29]	NiZn (β)	*B*2	2.9143	Ni_2_Zn_11_ (γ)	*D*8_2_	8.9228	2.0
Cu-Zn [30]	CuZn (β)	*A*2	2.9967	Cu_5_Zn_8_ (γ)	*D*8_2_	8.8690	1.3
Fe-Al [5]	Fe_5_Al_8_	*D*8_2_	8.9757	FeAl	*B*2	2.9720	0.7

## Data Availability

Data sharing is not applicable for this article.

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
