# Peer review of "The Mechanism of Dendrite Formation in a Solid-State Transformation of High Aluminum Fe-Al Alloys"

_materials, 2023, doi:10.3390/ma16072691_

Round 1

Reviewer 1 Report

The article is devoted to an interesting problem - the study of the mechanism of growth of dendrites of phases precipitating in hypoeutectoid alloys before the onset of eutectoid decomposition. The mechanism is considered on the example of alloys of the Fe (40.1 at%)-Al (59.9 at.%) system.

Available remarks:

1. The authors compare on page 3 the lattice parameters of Fe5Al8, FeAl and FeAl2, however, Table 1 shows data only for Fe5Al8 and FeAl.

2. The authors cooled the alloy in the decomposition zone of the hypoeutectoid alloy with two sharply different cooling rates (Scheme 1 and Scheme 2). According to known theoretical concepts, an increase in the cooling rate results in an increase in the area of the quasi-eutectoid and a decrease in the proportion of the excess phase (FeAl). Figures 3 and 4 do not confirm this relationship. The authors did not give an explanation for the deviation of the experimental results from the generally accepted.

3. I would like to see the structure of the dendrites formed during cooling according to scheme 1 and scheme 2 on a larger scale. This will make it possible to clearly distinguish between dendritic structures and eutectoid decomposition products.

4. The authors propose two necessary conditions for the formation of solid-state dendrites. These conditions are rather abstract. Their concretization is desirable, for example, the range of maximum mismatch of crystal structures or orientation ratios.

Author Response

Dear Reviewer:

Thank you for your careful review and professional comments concerning our manuscript entitled “The Mechanism of Dendrite Formation in a Solid-State Transformation of High Aluminum Fe-Al Alloy”. Those comments are all valuable and very helpful for revising and improving our paper, as well as the important guiding significance to our work. We have studied the comments carefully and have made a correction which we hope meets with approval. Revised portions are marked in red on the manuscript. The responses to the reviewer’s comments are as follows:

  1. The authors compare on page 3 the lattice parameters of Fe5Al8, FeAl and FeAl2, however, Table 1 shows data only for Fe5Al8 and FeAl.

Response:

Thank you for your professional comments.

Table 1 lists only those phases related to the formation of dendritic precipitates, i.e., the parent phase (Fe5Al8) and the phase showing dendritic morphology (B2 FeAl). As described in the manuscript, FeAl2 does not precipitate with a dendritic morphology (due to its complex crystal structure, which is incoherent with the cubic Fe5Al8 lattice), which is also why the entire manuscript focus on hypoeutectoid material (where the primary precipitates from Fe5Al8 are B2 FeAl). For convenience, we now have added the lattice parameters of FeAl2 (now included in the text on line 96-100). “….With the lattice parameter of Fe5Al8 being about three times that of FeAl and the lattice mismatch being only 0.7% (cf. Table 1), while the triclinic FeAl2 phase (P1, No. 2, aP19; a = 4.9548(3) Å, b = 6.5661(6) Å, c = 8.925(1) Å, α = 91.513(7)°, β = 73.446(7)°, γ = 97.058(7)° at 1080°C [5]) is completely incoherent with Fe5Al8.”

  1. The authors cooled the alloy in the decomposition zone of the hypoeutectoid alloy with two sharply different cooling rates (Scheme 1 and Scheme 2). According to known theoretical concepts, an increase in the cooling rate results in an increase in the area of the quasi-eutectoid and a decrease in the proportion of the excess phase (FeAl). Figures 3 and 4 do not confirm this relationship. The authors did not give an explanation for the deviation of the experimental results from the generally accepted.

Response:

Very good review. For the case of so-called proportional deviation of the phases, due to the limited magnification of the equipment used for in-situ observation, this paper focuses on understanding the preferential site of the nucleation as well as the size and morphological evolution of the proeutectoid FeAl phase, so it is chosen to capture the localization that is close to the grain boundaries at a higher magnification (scale of 25 um). In addition, diffusion will be faster on the free surface than that of in the bulk, Therefore, the proportion of the precipitated phase will have some deviation. For the growth of the solid-state phase after nucleation, it is usually controlled by a mixed mode (bulk diffusion-interfacial reaction). In the slow cooling case, the FeAl nuclei growth is mainly controlled by long-range diffusion. While in the rapid cooling case, the proeutectoid FeAl growth may be mainly controlled by rapid interfacial reactions (i.e. short-range diffusion), which results in higher volume fraction of proeutectoid phase than the slow cooling case. Similar unusual phenomenon has been observed in the literature[2]. At present, our results are not sufficient to give a definite explanation, but of course, this phenomenon will be worth studying in depth in the future. I recommend this discussion as an open comment.

  1. I would like to see the structure of the dendrites formed during cooling according to scheme 1 and scheme 2 on a larger scale. This will make it possible to clearly distinguish between dendritic structures and eutectoid decomposition products.

Response:

We have done a lot of in-situ experiments, unfortunately, the results are never as good as in the manuscript. Due to the limited magnification of the equipment for in-situ observation, the present magnification is already the highest (scale of 25 um). It is certain that the morphology of dendritic proeutectoid FeAl phase is very different from that of the lamellar eutectoid decomposition. In fact, the morphology of the lamellar eutectoid decomposition products of Fe5Al8 shows extremely thin size of about 200 nm (as shown in Fig. 9b in the manuscript).

  1. The authors propose two necessary conditions for the formation of solid-state dendrites. These conditions are rather abstract. Their concretization is desirable, for example, the range of maximum mismatch of crystal structures or orientation ratios.

Response:

The present results do not sufficient to give an upper limit for the lattice mismatch of the crystal structures or orientation ratios. However, the examples presented in Table 1 show that the formation of precipitates with dendritic morphology is possible at least up to a mismatch of 2% and the corresponding crystal structure can be cubic superstructure D82-B2, A1-L12, etc.[3]). I recommend this discussion as an open comment.

Thank you again for your comments and kind advice, your review comments were extremely helpful to improve our manuscript.

Sincerely,

Haodong Yang

References

  1. Stein, F.; Vogel, S.C.; Eumann, M.; Palm, M. Determination of the crystal structure of the É› phase in the Fe–Al system by high-temperature neutron diffraction. Intermetallics 2010, 18, 150-156, doi:https://doi.org/10.1016/j.intermet.2009.07.006.
  2. Contieri, R.J.; Lopes, E.S.N.; Caram, R.; Devaraj, A.; Nag, S.; Banerjee, R. Effects of cooling rate on the microstructure and solute partitioning in near eutectoid Ti–Cu alloys. Philosophical Magazine 2014, 94, 2350-2371, doi:https://doi.org/10.1080/14786435.2014.913113.
  3. Baumann, S.; Williams, D. Experimental observations on the nucleation and growth of δ′ (Al 3 Li) in dilute Al-Li alloys. Metallurgical and Materials Transactions A-physical Metallurgy and Materials Science - METALL MATER TRANS A 1985, 16, 1203-1211, doi:https://doi.org/10.1007/BF02670325.

Reviewer 2 Report

This paper reports on the mechanism of dendrite formation in a solid-state transformation of high Al content Fe-Al alloy. The paper is well written nd the results are presented and discussed well. Some minor changes can be made to improve the quality:

1) The introduction can describe in some more detail the need to examine these high Al content alloys and also the implications of the dendrite precipitations to the overall performance of the material. 

2) Materials and methods - what about any impurities in the studied Fe-Al alloys? Could they have impacted the microstructural changes? 

3) Materials and methods - why was the selected profile chosen? 

4) Conclusions - What are the implications of this phase precipitation on the strength and oxidation resistance of the material? 

Author Response

Dear Reviewer:

Thank you for your careful review and professional comments concerning our manuscript entitled “The Mechanism of Dendrite Formation in a Solid-State Transformation of High Aluminum Fe-Al Alloy”. Those comments are all valuable and important guiding significance to our work. We have studied the comments carefully and our responses are as follows:

  • The introduction can describe in some more detail the need to examine these high Al content alloys and also the implications of the dendrite precipitations to the overall performance of the material.

Response:

Thank you for your professional comments.

This aspect was also considered previously in our earlier version, however, the purpose of this paper is to perform a fundamental study on the formation mechanism of special solid-state dendrites, so the performance effects and discussion are not addressed. Of course, as a theoretical study, this is not sufficient. In the future, our research will explore the effect of dendrite morphology on the performance of material. So, we decided the introduction should not be revised.

  • Materials and methods - what about any impurities in the studied Fe-Al alloys? Could they have impacted the microstructural changes?

Response:

The Fe-Al binary alloy was produced by vacuum induction melting using high-purity Fe (99.95 wt.% purity) and Al (99.999 wt.% purity). EPMA analysis did not show indications of impurities, which were negligible in extremely trace amounts. Of course, in the high-temperature in-situ experiments, small oxide particles inevitably form on the sample surface (This is also described in the manuscript). Nevertheless, the original observation of this phenomenon of solid-state dendrite formation (published in a preceding paper [5] by one of the authors) was in bulk material, i.e., possibly existing impurities on the surface are not the reason for the observed dendrite morphology.

  • Materials and methods - why was the selected profile chosen?

Response:

I guess that the “profile” refers to the cross section selected by the EBSD analysis. The selected profile is characterized by the surface with the largest cross-sectional area of the dendrites (preferential growth plane) determined after polishing at an adjustable angle.

  • Conclusions - What are the implications of this phase precipitation on the strength and oxidation resistance of the material?

Response:

Similar to the first answer. The scope of our investigations was focused on understanding the mechanism of the FeAl dendritic formation. It is just a foundamental research. Therefore, effects on the strength and oxidation behavior were not studied in the present project. So, we decided the introduction should not be revised.

Sincerely,

Haodong Yang
